# Moderating role of observing the five precepts of Buddhism on neuroticism, perceived stress, and depressive symptoms

Nahathai Wongpakaran[1], Phurich Pooriwarangkakul[2], Nadnipa Suwannachot[2], Zsuzsanna Mirnics[3], Zsuzsanna Kövi[3]*, Tinakon Wongpakaran[1]*

1 Faculty of Medicine, Department of Psychiatry, Chiang Mai University, Chiang Mai, Thailand, 2 Faculty of Medicine, Chiang Mai University, Chiang Mai, Thailand, 3 Károli Gáspár University of the Reformed Church, Budapest, Hungary

* tinakon.w@cmu.ac.th (TW); kovi.zsuzsanna@kre.hu (ZK)

**Data Availability Statement:** All relevant data are within the paper and its Supporting Information files.

## Abstract

### Purpose

Evidence has shown that the Five precepts significantly affect the relationship between attachment and resilience; however, little is known whether observing the Five Precepts would help reduce depressive symptoms among those who experience risks. The aim of this study was to examine the moderating role of the Five Precepts in the mediation model relationship among neuroticism, perceived stress, and depression.

### Patients and methods

The study employed a cross-sectional survey design and data were collected from the end of 2019 to September 2022 in Thailand. In all, 644 general participants completed questionnaires on the Neuroticism Inventory (NI), the 10-item Perceived Stress Scale (PSS), Depression Subscale, and the Five-Precept Subscale of the Inner Strength-based Inventory (SBI-PP). Mediation and moderation analyses with 5000 bootstrapping methods were used.

### Results

Among all, 74.2% were female, and the mean age totaled 28.28 years (SD = 10.6). SBI-PP was shown to have a moderation effect on the relationship between NI, PSS and depressive symptoms. The moderating effect between SBI-PP and PSS was significant, whereas SBI-PP and NI was not. The index of moderated mediation from the Five Precepts was significant (b = -0.019 (95%CI -0.029, -0.009)). The moderated mediation model increased the percent variance explaining depressive symptoms to 47.6%, compared with 32.6% from the mediation model alone.

### Conclusion

Observing the Five Precepts offers evidence that it buffers the effect of perceived stress on depression. People with high levels of observing the Five Precepts are less likely to develop depressive symptoms. Implications as well as possible future research are discussed.

**Funding:** This research was supported by the Faculty of Medicine Research Fund of Chiang Mai University (grant no. 152/2562). The funders had no role in study design, data collection and analysis, decision to publish, or preparation of the manuscript.

**Competing interests:** The authors have declared that no competing interests exist.

## Introduction

Around 322 million people in this world are living with depression, and nearly one half of these people live in the Southeast Asia and Western Pacific Regions [1]. Many factors are related to the development of depression, and one of those is the personality trait of neuroticism [2–5]. Neuroticism is one of the Big Five personality dimensions, characterized by the tendency to experience negative emotions, including anger, anxiety, fear, self-consciousness, irritability, emotional instability, and depression [6,7]. A clinically significant depressive symptom is usually attributable to an interaction of the trait of neuroticism with a life stressor [7].

Literature reviews show that the relationship between neuroticism and depression is robust, with the risk ratio of 1.25 (95% CI: 1.04, 1.45) [8]. This significant relationship is also evident across cultures [2–5,9].

In addition to the direct effect on depression, neuroticism is found to have an indirect effect through other variables such as social inhibition [4,10], and perceived stress [11], catastrophic and anxiety-provoking appraisals [12], and cognitive emotion regulation [13]. In particular, perceived stress, thoughts or the feelings that individuals experience after encountering stressful life events, is strongly associated and often significantly antecedent to depression, which has been shown to be a mediator or moderator of the effect of neuroticism on depression [14–24].

The relationship among neuroticism, perceived stress, and depression is illustrated in many related research articles [3,4,25–29]. Evidence has shown that neuroticism indirectly affects depression through perceived stress.

On the other hand, the effect of neuroticism and perceived stress on depression may be buffered by the positive variables involved [30,31], including self-efficacy [32,33], resilience [34], equanimity [35], and the religious participation [36]. Relating to equanimity, a strength found in Buddhist discipline and the one the authors have found relevant in clinical encounters is the observance of the Five Precepts. The Five Precepts are one of the most well-known and common practices for Buddhists. The Five Precepts include refraining from killing, stealing, sexual misconduct, telling bad-intentioned lies, and using intoxicants [37].

The observance of precepts (*Sila*) serves as the preliminary foundation to refine higher virtues development, and the most important step on the spiritual journey. Paving the way to right concentration, and wisdom is considered a crucial practice, that would lead the person to the highest religious goal of *Nibbana* [38,39]. For general people or nonserious practitioners, observance of the Five Precepts increases wellbeing and quality of life [40,41]. However, observance of the Five Precepts is not well-known among international academic circles compared with mindfulness meditation, despite the fact that thousands of articles of observance of the Five Precepts have been published in Thailand [42].

By its characteristics, precepts may be seen as socially adaptive behaviors requiring motivation, and self-control to carry out. As the Five Precepts include items related to congruent moral behavior, higher values on this scale might mean higher self-congruence (together with higher self-control), and this can be a part of the buffering effect of perceived stress on depression. While self-control is shown to have a moderating effect on positive outcomes such as self-efficacy, academic success [43], and self-management behaviors [44]. Little is known about the role of observance of the Five Precepts on negative mental outcomes such as perceived stress, neuroticism and depression. The authors therefore analyzed to see whether observance of the Five Precepts would serve as a buffer for any mental health outcome the same way as self-control does. Specifically, the authors examine the moderating effect on the relationship among neuroticism, perceived stress and depression. We hypothesized that precept practice may buffer the relationship between neuroticism, perceived stress, and

depression. By that the high level of precept observance would reduce the effect of neuroticism and perceived stress on depressive symptom.

## Material and methods

### Participants

This study was conducted using an online survey in Thailand from December 2019 to September 2020. The target group comprised the general population. A convenience sampling method was applied. Flyers, websites, Facebook, Instagram, and LINE were used to invite participation. Inclusion criteria included 1) age between 18 and 59 years, 2) fluent in Thai and 3) able to access to the Internet and Google form. Exclusion criteria consisted of 1) having psychiatric history or being treated for psychiatric disorder and 2) being intoxicated.

Sample size estimation for power analysis of the mediation model was based on correlation coefficients between variables from the prior result, and type I error (alpha) at 0.05, type II error (beta, 1-power) at 0.01, with two-tailed test of significance. The expected minimum sample size was 95 to yield a power of 99% (95%CI .95, 1.00). However, in this survey 644 respondents took part in the study, and we used all data for analysis. Each gave written informed consent before filling out the questionnaires. The study was conducted according to the guidelines of the Declaration of Helsinki and approved by the Institutional Review Board (or Ethics Committee) of the Faculty of Medicine, Chiang Mai University (study code, 184/2562 and date of approval, 8 July 2019).

### Measurements

**10-item Perceived Stress Scale (PSS-10).** This scale is used to assess to what extent the respondent feels about the stress he/she has perceived over four weeks. PSS-10 comprises 10 items and uses a 5-point Likert type scale format (0 = never to 4 = very often), and the total score ranges from 0 to 40 [45]. Higher scores suggest greater perceived stress. PSS-10 has been widely used for both clinical and nonclinical samples. The Thai version showed good reliability and validity [46]. In this study sample, the PSS-10 demonstrated a good internal consistency (Cronbach's alpha was .78).

**Neuroticism Inventory (NI).** The NI is a dimensional measure of the neuroticism personality trait based on Eysenck's five-factor model [47]. The NI, developed by Wongpakaran et al., consists of a self-rating scale including 15 items with a 0 to 4 Likert type scale (1 = never like me to 4 = always like me) [48]. and the total score ranges from 15 to 60. A higher score indicates a higher level of neuroticism. The previous studies showed that the NI had high internal consistency (Cronbach's alpha was .91 - .92) [35, 49],and NI showed good validity and reliability [48]. In this study sample, the Cronbach's alpha was .90.

**Core Symptom Index -Depression subscale (CSI-D).** CSI is a scale used to measure common psychological symptoms. The CSI instructions asked respondents to answer the items based on how they felt over the past week [50]. The CSI consisted of 17 items, 5 items representing depression, 4 items for anxiety, and 6 items for somatization symptoms. Response options were based on a 5-point Likert type scale, i.e., values of 0 (never), 1 (rarely), 2 (sometimes), 3 (frequently) and 4 (almost always), and the total score ranges from 0 to 60. The higher the score reflects the higher the level of psychopathology. The CSI showed good validity and reliability [51]. Depression subscale (CSI-D) was used in this study, and the total score ranges from 0 to 20. In this study sample, the CSI-D demonstrated a good internal consistency (Cronbach's alpha was .79).

**Precept Practice (SBI-PP) or Observance Five Precepts.** SBI-PP is an item drawn from the 10 inner Strength-Based Inventory (SBI), e.g., loving-kindness and equanimity [52]. It

comprises a single item with 5 multiple-choice options. The SBI item provided optional outcome response attributing to the cognitive-behavioral aspect of each strength. Precept practice is to measure the level of observing the Five Precepts. The stem begins with moral virtues including to refrain from 1) killing, 2) stealing, 3) sexual misconduct, 4) telling bad-intentioned lies, and 5) intoxicants such as alcohol and addictive drugs". The response choices ranged from 1 "I never thought to follow the moral virtues" to 5, "I always follow the moral virtues. As I can remember, I have never broken them before". A higher score indicates a higher level of observance of the Five Precepts. SBI-PP was significantly correlated with other strengths, e.g., patience and endurance, r = .164, p < .001). As SBI-PP is a single item, internal consistency is not calculated. Nevertheless, test-retest may be a better option for evaluating the participant consistency [53]. Two-week test-retest reliability using intraclass correlation coefficient of the SBI-PP was .87 (95%CI = .70, .95, p < .0001), indicating good reliability.

## Statistical analysis

Descriptive statistics was used for sociodemographic and scores of the measurements. Mean and standard deviation were calculated for continuous data, i.e., the total score of each measurement. Correlation analysis for continuous variables, e.g., CSI-dep and PSS used Pearson's correlation, for categorical or ordinal variables, e.g., sex and education, polychoric correlation was used, for categorical or ordinal and continuous variables, e.g., marital status and neuroticism, polyserial correlation was performed to determine significant relationships between variables.

Data were checked and shown to have normal error distribution, linearity and homoscedasticity. No multicollinearity and outliers were demonstrated. All indicated valid data for performing mediation analysis. Mediation and moderation analyses were carried out, beginning with testing the mediation model of neuroticism, depression, and perceived stress. By that neuroticism was independent (X) and depression was outcome (Y), whereas perceived stress served as a mediator (M1) [54]. The path or regression coefficients between X and M, M and Y, and X and Y was a, b, and c', respectively (Fig 1).

For moderation analysis, the plots were created between neuroticism (X) and perceived stress (M), neuroticism (X) and depression (Y), and between perceived stress (M) and depression (Y), according to the high and low levels of observing the Five Precepts. Significant interaction of each plot was investigated by visualizing predicted values of neuroticism or perceived stress scores with the high or low level of observing the Five Precepts [54]. The moderation model that illustrated the presence of moderating effects would be included in the full moderated moderation model. According to Hayes [54], if the moderation effect existed at a, b, and c', then 7 moderated mediation models were possibly produced; and therefore, each model would be tested.

To produce more accurate results of mediation and moderation analysis, resampling or bootstrapping methods was applied [54,55]. The results were reported by unstandardized estimates, standard errors, p-values. Bootstrap confidence intervals for conditional indirect effects were applied. We used bootstrap instead of traditional Baron and Kenny's mediation methods and the Sobel test because indirect effects are unlikely to be normally distributed. Bootstrap is a resampling technique with replacement, and no assumption is made about the shape of the sampling distribution of indirect effect, the results will get more credibility, and the bootstrap confidence interval tends to have higher power than the Sobel test [56]. Confidence intervals that do not include zero are indicative of statistical significance. For all the analyses, the level of significance was set at $p$ <0.05. All statistical analyses were carried out using the IBM SPSS Program, 22.0. MedCalc, Version 19.7 was used to produce scatter plots and regression lines.

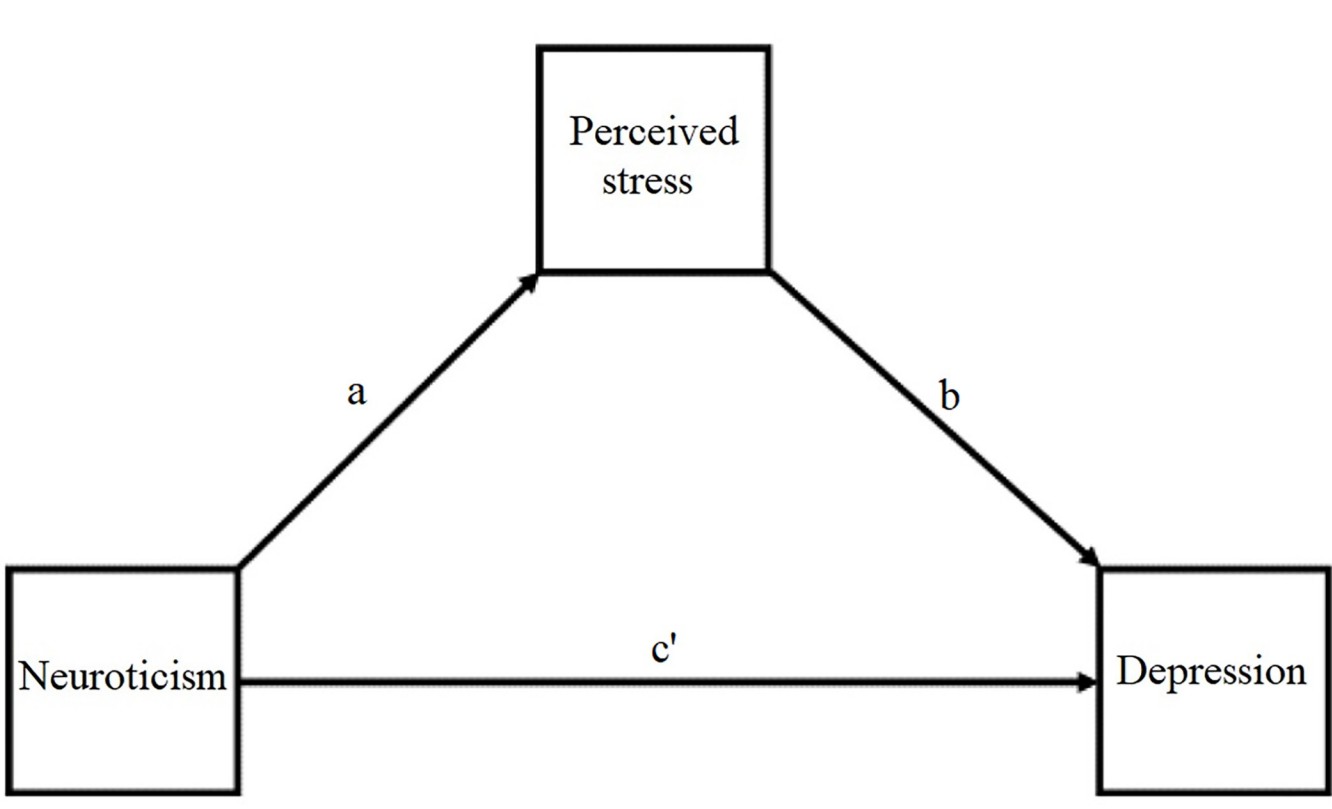

**Fig 1. The hypothesized mediation model.**

PROCESS, Version 3.5 annexed to IBM SPSS was used for all mediation and moderation analyses.

## Results

The participants' ages ranged from 18 to 72, with an average of 28 years old. Over 70 were female, lived alone, and had obtained a bachelor's degree. Over half of the participants earned a moderate level of income. All participants were Thai, and 93.3% were Buddhist. For clinical variables, the average NI, CSI-D, and PSS scores were mild to moderate, whereas the SBI-PP score of the sample was slightly over the mid-point. The details are shown in Table 1.

Table 2 shows the correlation coefficients among variables. Being male was associated with higher education (p < .05) but negatively related to the low level of income (p < .05) and the level of SBI-PP(p < .01). Age was associated with living alone, high level of education, income, and high level of SBI-PP (all p < .01), but negatively related to CSI-D score (p < .01). Living alone was positively associated with the level of education and income (all p < .01), but not with other clinical variables. Monthly income was negatively associated only with CSI-D (p < .05). As expected, NI was positively correlated with CSI-D and PSS scores, but negatively associated with SBI-PP score (all p < .01). CSI-D and PSS scores were positively correlated but negatively correlated with SBI-PP (p < .01). Along the same line, PSS was conversely related to SBI-PP scores(p < .01).

Hierarchical regression analysis was used to identify potential confounders. The results showed that age, sex, and marital status were the significant predictors that reduced the effect

**Table 1. Socio–demographic characteristics of the participants (n = 644).**

| Variable | | Value |
|---|---|---|
| Sex | n (%) | |
| | Female | 478 (74.2) |
| | Male | 166 (25.8) |
| Age | M±SD | 28.28 ± 10.6 |
| Living status | n (%) | |
| | Lived alone | 519 (80.6) |
| | Lived wih partner | 125 (19.4) |
| Monthly income | n (%) | |
| | ≤ 20,000 THB | 405 (62.9) |
| | 20,000 THB and more | 239 (37.1) |
| Education | n (%) | |
| | lower than Bachelor | 67 (10.4) |
| | Bachelor | 456 (70.8) |
| | Master and higher | 121 (18.8) |
| Clinical variable | M±SD | |
| | NI score | 33.40 ± 9.0 |
| | CSI-D score | 3.05 ± 3.09 |
| | PSS score | 15.08 ± 6.0 |
| | SBI-PP score | 3.34 ± 1.03 |

Notes

NI = Neuroticism Inventory, CSI–D = Depression scale of Core Symptom Index, PSS = Perceived stress scale, SBI–PP = Equanimity scale of the inner Strength–based Inventory, THB = Thai baht; 1 THB = 0.026 US Dollars, M = mean, SD = standard deviation.

size of neuroticism on depression, which was considered confounders; therefore, these three variables were controlled as covariates in the moderated mediation model.

Table 3 shows the summary of mediation analysis of neuroticism and perceived stress predicting depressive symptoms controlling for age, sex, and marital status. NI, PSS, and marital status (lived alone) predicted depressive symptoms (t = 11.000, p < .0001, t = 10.350, p < .0001, and t = 4.334, p < .0001, respectively.). By adding PSS, the model of the variance of

**Table 2. Correlation matrix among variables.**

| | 1 | 2 | 3 | 4 | 5 | 6 | 7 | 8 |
|---|---|---|---|---|---|---|---|---|
| 1.Sex, male | | | | | | | | |
| 2.Age | .009 | | | | | | | |
| 3. Living, lived with partner | -.053 | -.602** | | | | | | |
| 4.Education, bachelor | .197* | .267** | -.288** | | | | | |
| 5.Monthly Income, <700 US | -.194* | .668** | -.453** | .485** | | | | |
| 6. NI score | .030 | .002 | -.012 | -.066 | .029 | | | |
| 7. CSI-D score | -.026 | -.166** | .168** | -.062 | -.090* | .578** | | |
| 8. PSS score | .004 | -.062 | .001 | -.062 | -.054 | .599** | .611** | |
| 9. SBI-PP score | -.126** | .128** | -.123** | .049 | .013 | -.215** | -.236** | -.148** |

*p< 0.05

**p< 0.01, NI = Neuroticism Inventory, CSI–D = Depression scale of Core Symptom Index, PSS = Perceived stress scale, SBI–PP = Precept practice scale of the inner Strength–based Inventory.

**Table 3. Summary of mediation analysis of neuroticism and perceived stress predicting depressive symptoms controlling for age, sex, and living status.**

|  |  | M(PSS) |  |  | Y(CSI-Dep) |  |  |
| --- | --- | --- | --- | --- | --- | --- | --- |
| Antecedent |  | Coeff. | SE | p-value | Coeff. | SE | p-value |
| X(NI) |  | 0.352 | 0.020 | .000 | 0.132 | 0.012 | .004 |
| M(PSS) |  | - | - | - | 0.207 | 0.020 | .000 |
| age |  | -0.037 | 0.021 | .073 | -0.014 | 0.011 | .175 |
| sex |  | -0.302 | 0.404 | .456 | -0.370 | 0.207 | .076 |
| Living status, lived with partner |  | 0.912 | 0.557 | .102 | -1.244 | 0.287 | .000 |
| Constant |  | 4.122 | 0.877 | .000 | -3.653 | 0.458 | .000 |
|  | $R^2 = .326$ |  |  |  | $R^2 = .455$ |  |  |
|  | $F_{(4,639)} = 77.290, p < .0001$ |  |  |  | $F_{(5,638)} = 106.708, p < .0001$ |  |  |

Note: NI = Neuroticism Inventory, CSI–D = Depression scale of Core Symptom Index, PSS = Perceived stress scale, SBI–PP = SBI–PP = Practice precept scale of the inner Strength–based Inventory, SE = standard error.

depressive symptom increased from 36.3% to 45.5%. NI had a significantly indirect effect via PSS ($\beta = .072$, $p < .001$).

Fig 2 displays the slope of the regression lines along with the observation between PSS and CSI-D. In the low practice level of the Five Precepts, the slope coefficient was .375 ($p < .001$), whereas in the high-level practice population, the slope coefficient was .244 ($p < .001$). A significant difference between two slopes was noted (t = -3.561, $p < .001$).

Fig 3 displays the slope of the regression line along with the observation between NI and CSI-D. In the low practice level of the Five Precepts, the slope coefficient was .225 ($p < .001$),

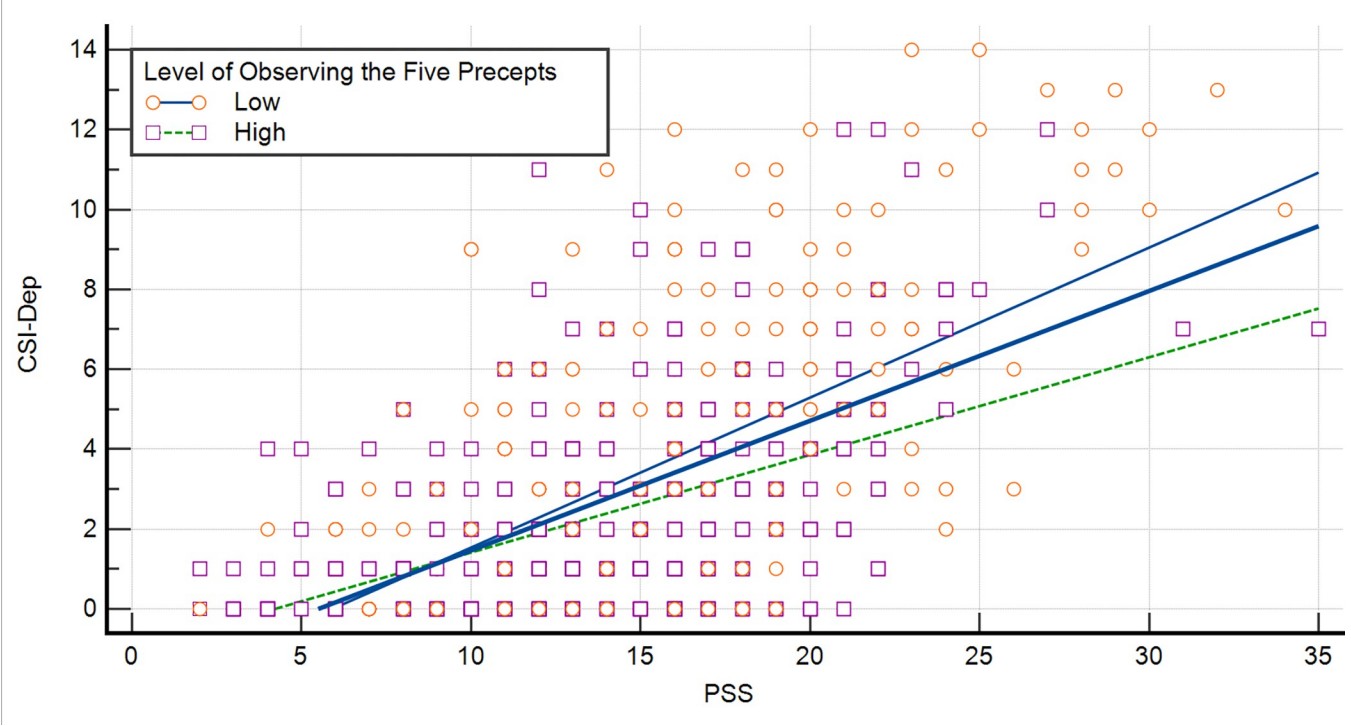

**Fig 2. Regression lines between CSI–D and PSS scores based on the level of SBI–PP.**

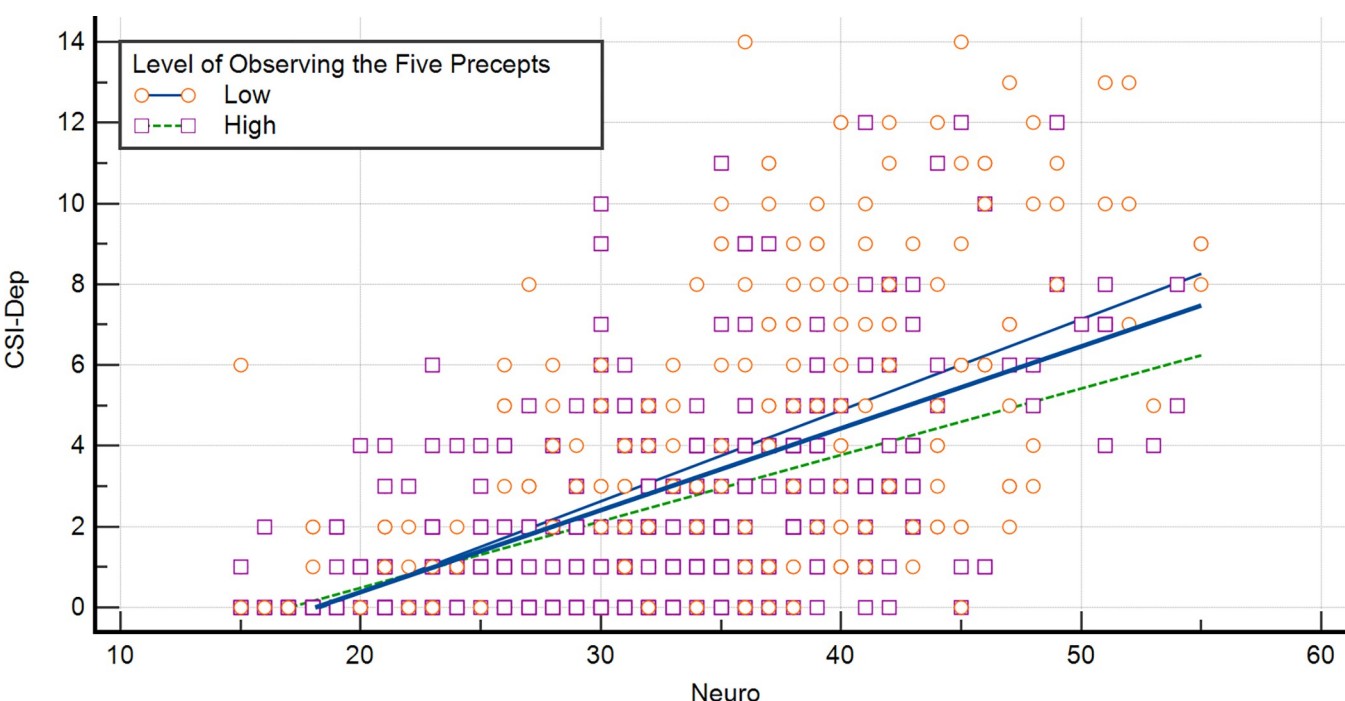

**Fig 3. Regression lines between CSI–D and NI scores based on the level of SBI–PP.**

whereas in the high-level practice population, the slope coefficient was .164 (p < .001). A significant difference between two slopes was noted (t = -2.644, p = .008).

For the regression lines between PSS and NI, the slope coefficient was 0.361 (p < .001) in the low practice level of the Five Precepts, and the slope coefficient was 0.335 (p < .001) in the high-level practice population. However, no significant difference was observed between the two slopes (t = -.620, p = .535) (Figure not shown).

Based on the significant interaction effect between SBI-PP and PSS, but not SBI-PP and NI, the two possible models were Models 14 and Model 15. Model 14 indicated the moderation of the relationship between PSS and CSI-D by SBI-PP (b path), whereas Model 15 illustrated the moderating effect of SBI-PP between PSS and CSI-D, and between NI and CSI-Dep (c' path) (Fig 4). The results showed that Model 14 best described the data. The variance of CSI-D was explained by this model for 47.6%.

Table 4 shows the significant direct and indirect effects of the predictors. The moderating effect of SBI-PP and PSS was shown to be negatively associated with depression scores. This could be interpreted in that for the low level of precepts practice, every score of stress perceived provides us 0.273 point on depression score. For the average level precepts practice, every score of stress perceived provides us 0.215 point on depression score, and for the high level of precepts practice, every score of stress perceived provides us 0.157 point on depression score. The index of moderated mediation model was significant (B = -0.021, 95% CI: -.033, -0.009). That is, the mediation of the effect of neuroticism on depression through perceived stress is moderated by the precepts practice.

Table 4 shows the summary of moderated mediation analysis of precept practice, neuroticism and perceived stress predicting depressive symptoms controlling for age, sex, and marital status. NI, PSS, sex, and marital status (being alone) predicted the depressive symptoms (t = 2.95, p = .004, t = 6.29, p < .0001, t = 2.16, p = .030, and t = 4.334, p = .0001, respectively.). SBI-PP showed a moderating effect only on PSS but not on NI (t = 3.22, p = .001). By adding

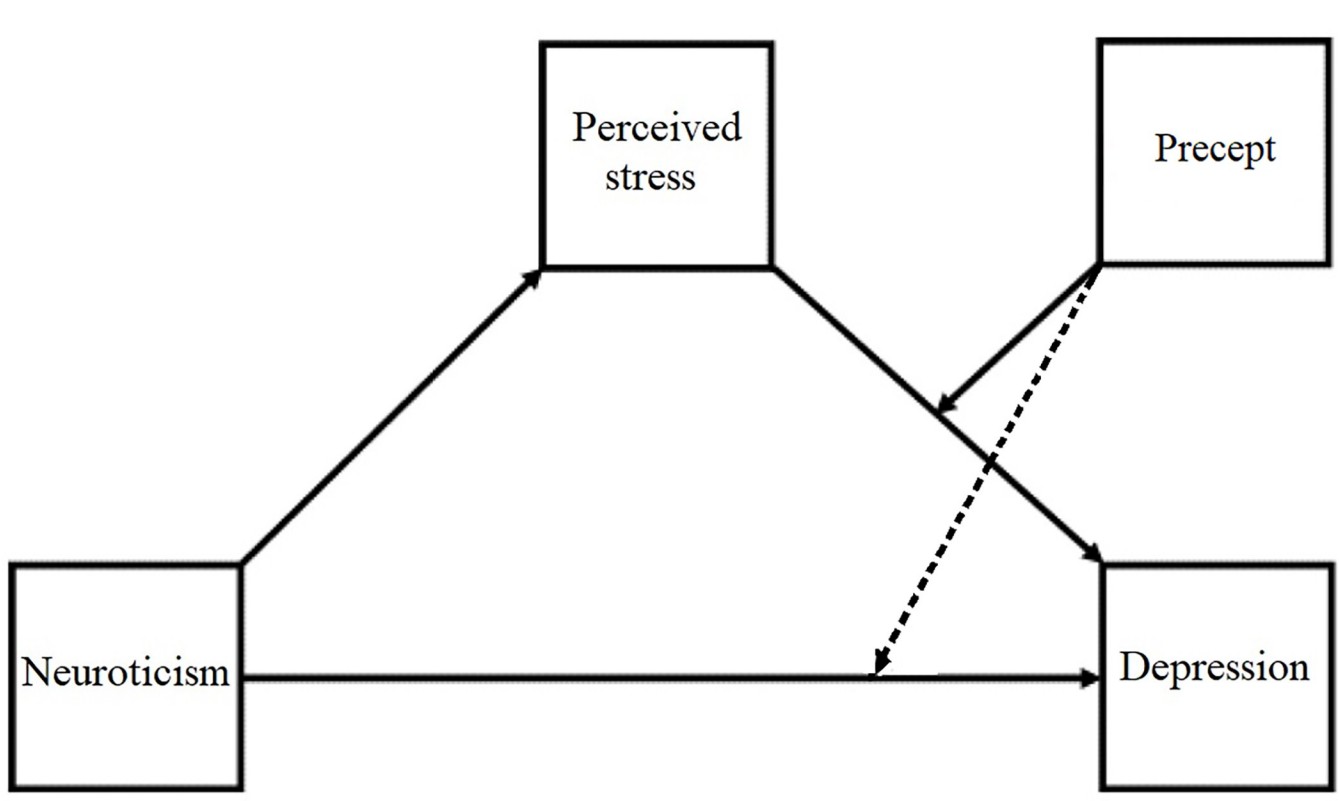

**Fig 4. Possible moderated mediation models (Model 14 and Model 15).** Legend: Model 14 (excluding the dotted line) and Model 15 (including the dotted line).

**Table 4. Conditional indirect effect of neuroticism on depressive symptoms at values of the moderator observance of Five the Precepts through perceived stress.**

| | | Consequent | | | | | | |
|---|---|---|---|---|---|---|---|---|
| | | M(PSS) | | | | Y(CSI-Dep) | | |
| Antecedent | | Coeff. | SE | p-value | | Coeff. | SE | p-value |
| X(NI) | | 0.352 | 0.020 | .000 | | 0.127 | 0.043 | .004 |
| M(PSS) | | | | | | 0.390 | 0.062 | .000 |
| W(SBI-PP) | | | | | | 0.634 | 0.351 | .072 |
| X*W | | | | | | 0.000 | 0.012 | .975 |
| M*W | | | | | | -0.058 | 0.018 | .001 |
| age | | -0.037 | 0.021 | .073 | | -0.016 | 0.011 | .138 |
| sex | | -0.302 | 0.404 | .456 | | -0.448 | 0.207 | .030 |
| Living status, lived with partner | | 0.912 | 0.557 | .102 | | -1.144 | 0.283 | .000 |
| Constant | | 4.122 | 0.877 | .000 | | -5.473 | 1.284 | .000 |
| | $R^2 = .326$ | | | | | $R^2 = .476$ | | |
| | $F(4,639) = 77.290, p < .0001$ | | | | | $F(8,635) = 72.209, p < .0001$ | | |

Note: NI = Neuroticism Inventory, CSI–D = Depression scale of Core Symptom Index, PSS = Perceived stress scale, SBI–PP = Practice precept scale of the inner Strength–based Inventory, SE = standard error.

**Table 5. The conditional direct and indirect effects of X on Y.**

*Conditional direct effect(s) of X on Y:*

| SBI-PP | Effect | SE | t | p-value | LLCI | ULCI |
|---|---|---|---|---|---|---|
| 2 (low) | .128 | .021 | 5.979 | .000 | .086 | .169 |
| 3 (average) | .128 | .014 | 9.476 | .000 | .101 | .154 |
| 4 (high) | .128 | .015 | 8.791 | .000 | .100 | .157 |

*Conditional indirect effects of X on Y:*

Indirect effect: NI—> PSS—> CSI-Dep

| SBI-PP | Effect | BootSE | BootLLCI | BootULCI | | |
|---|---|---|---|---|---|---|
| 2 (low) | .096 | .013 | .072 | .123 | | |
| 3 (average) | .078 | .009 | .058 | .095 | | |
| 4 (high) | .055 | .009 | .040 | .073 | | |

*Index of moderated mediation of SBI-PP*

| Index | BootSE | BootLLCI | BootULCI | | | |
|---|---|---|---|---|---|---|
| -.021 | .006 | -.032 | -.009 | | | |

NI = Neuroticism Inventory, CSI–D = Depression scale of Core Symptom Index, PSS = Perceived stress scale, SBI–PP = Practice precept scale of the inner Strength–based Inventory, SE = standard error, BootSE = bootstrap standard error, BootULCI = bootstrap upper–level confidence interval, BootLLCI = bootstrap lower–level confidence interval.

interactions, the model explained the variance of depressive symptom more, from 36.3 to 47.6%. The index of moderated mediation model was significant (B = -0.021, 95% CI: -.032, -0.009).

Table 5 shows the direct effect of neuroticism → depression path was significant in all three conditions of the observance of the Five Precepts (low, medium, high). The conditional indirect effect of neuroticism → perceived stress → depression path was significant in all three conditions of the value of the observance of the Five Precepts; however, the conditional indirect effect decreased when the level of observance of the Five Precepts increased, denoted by the negative index of the moderated mediation (-.0206).

## Discussion

The present study examined the role of perceived stress on the relationship between neuroticism and depressive symptoms. Also, how observing the Five Precepts buffered their effects on depressive symptoms. The findings can be interpreted in that the effect of perceived stress on depression depended on the level of observing the Five Precepts. At a high level, the relationship between perceived stress and depression was significant lower. In all, observing the Five Precepts significantly buffered the perception of stress on depression.

As hypothesized, observing the Five Precepts can be viewed as behavioral control requiring many positive attributes to achieve. Five Precepts are not only a part of the ten perfections, but also viewed as a constitution of right speech, right livelihood, right action, the three of the Noble eightfold path, the principal teaching of Buddhism [57]. As mentioned, any attribute is not a standalone. Moral virtue requires a person to have right view, right effort, or right mindfulness for successful observance. This implies that a person who is practicing observing the Five Precepts may have elevated levels of their positive mental strength during such periods.

Even though no study has been reported before regarding this association, a comparable research could be discovered in resilience and equanimity that were shown to have moderating and mediating roles in the connection of neuroticism and depressive symptoms [34,35]. The mechanism of change of observing the Five Precepts may be similar to equanimity. It might be

involved in rendering a calming state of mind and living, and gaining more self-awareness, which would reduce the feeling of stress one is experiencing. One study revealed that the interaction of Buddhist affiliation and religious participation is negatively associated with depressive symptoms [36]. Observing Five Precepts might be a part of such religious participation and practice. However, more research is needed for a full explanation.

Like resilience or equanimity, observing the Five Precepts is a positive attribute that can be learned or acquired, while neuroticism is a trait that is more likely to be difficult to change. Cultivating the observance of the Five Precepts may change the association between neuroticism, perceived stress, and depressive symptoms.

Observing the Five Precepts should be encouraged to practice as mindfulness meditation. Based on Buddhism, it has been suggested to be practiced simultaneously. Even though, the Five Precepts, is from Buddhist ideology, non-Buddhists may adhere to this observance as this self- control behavior seem to make individuals adhering to it be regarded as a 'no harm and safe' person for society. It would be interesting to study this issue in a non-Buddhist culture.

Our findings suggested that people exhibiting high levels of neuroticism, and high levels of stress, may tend to develop depressive symptoms that may be buffered when obtaining a high level of observing the Five Precepts.

## Implications of the study

In clinical implication, observing the Five Precepts may be promoted along with any form of mindfulness meditation or mindfulness-related therapy [58–60]. In addition to buffering adverse mental health outcomes, observing the Five Precepts has been shown to be associated with well-being [61]. Therefore, it should be promoted even among the general population and those who have yet to experience stress. Researchers should carry out the practice of the Five Precepts further in the future. For example, research concerning an association between observing the Five Precepts and other positive strengths, such as resilience, grit, perseverance, and patience, should be examined, in addition to adverse mental health outcomes.

However, although Five Precepts can be viewed as healthy behaviors to be fostered for oneself and others, some, especially non-Buddhists, may find it uncomfortable when considering it as culture or religion related. Therefore, mental health professionals may adopt a careful approach emphasizing "behaviors" rather than religious matters, the same way mindfulness meditation is recognized. Such an approach may make it more acceptable and open to practice and further research.

## Strengths and limitations

Although, this study constitutes one of the first studies to assess a relationship among observing the Five Precepts, neuroticism, perceived stress, and depression, it encountered limitations. First, due to a cross-sectional design, any cause-effect relationships cannot be confirmed. Longitudinal data analysis should be warranted.

Second, this study was limited to people who could access the online survey. The invitation was carried out using social media and flyers. It was difficult to control for equality of sex and other demographic factors. The results can only tell that females and people who live alone participated the most in the study. The disproportionate sex ratio makes it unlikely to be representative of Thai people. Third, observing the Five Precepts is a single item presenting the levels of all Five Precepts as a whole. This might influence responses from those who adhere only to some precepts, so a separate five-item questionnaire in further research should help remove this doubt. Finally, we have no data about the religious involvement of the sample, therefore relation toward the Five Precepts is more an attitudinal and less a behavioral index, it is

unknown whether its subscales are correlated with religious practice. The present findings need to be supported by further behavioral indices of religious involvement in the future to better understand the meaning of the current results.

## Conclusion

Observing the Five Precepts showed evidence that it buffers the effect of perceived stress on depression. People with the high levels of observing the Five Precepts would be less likely to develop depressive symptoms. Implications for either clinical or nonclinical settings are discussed. Further research should be warranted.

## Supporting information

**S1 Dataset.**
(XLSX)

## Acknowledgments

We are thankful to our assistants for the data collection and all participants who made this research successful.

## Author Contributions

**Conceptualization:** Nahathai Wongpakaran, Phurich Pooriwarangkakul, Nadnipa Suwanna-chot, Zsuzsanna Mirnics, Zsuzsanna Kövi, Tinakon Wongpakaran.

**Data curation:** Phurich Pooriwarangkakul, Nadnipa Suwannachot, Tinakon Wongpakaran.

**Formal analysis:** Zsuzsanna Kövi, Tinakon Wongpakaran.

**Funding acquisition:** Tinakon Wongpakaran.

**Investigation:** Nahathai Wongpakaran, Phurich Pooriwarangkakul, Nadnipa Suwannachot, Zsuzsanna Mirnics, Zsuzsanna Kövi.

**Methodology:** Nahathai Wongpakaran, Zsuzsanna Kövi, Tinakon Wongpakaran.

**Project administration:** Nahathai Wongpakaran.

**Resources:** Nahathai Wongpakaran.

**Software:** Tinakon Wongpakaran.

**Supervision:** Nahathai Wongpakaran.

**Validation:** Nahathai Wongpakaran, Zsuzsanna Mirnics, Zsuzsanna Kövi.

**Visualization:** Phurich Pooriwarangkakul, Nadnipa Suwannachot.

**Writing – original draft:** Nahathai Wongpakaran, Phurich Pooriwarangkakul, Nadnipa Suwannachot, Zsuzsanna Mirnics, Zsuzsanna Kövi, Tinakon Wongpakaran.

**Writing – review & editing:** Nahathai Wongpakaran, Phurich Pooriwarangkakul, Nadnipa Suwannachot, Zsuzsanna Mirnics, Zsuzsanna Kövi, Tinakon Wongpakaran.

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
