## [Decision Letter · Decision Letter 0]

15 Sep 2022

PONE-D-22-01483Moderating role of observing the five precepts on neuroticism, perceived stress, and depressive symptomsPLOS ONE

Dear Dr. Wongpakaran,

Thank you for submitting your manuscript to PLOS ONE. After careful consideration, we feel that it has merit but does not fully meet PLOS ONE’s publication criteria as it currently stands. Therefore, we invite you to submit a revised version of the manuscript that addresses the points raised during the review process. Two reviewers have now evaluated your submission. Both have identified a number of opportunities to improve the manuscript, including by clarifying aspects of the study design and methods and modifying the presentation of the results. Please respond carefully to all of the points the reviewers have raised when preparing your revision.

We look forward to receiving your revised manuscript.

Kind regards,

Jamie Males

Editorial Office

PLOS ONE

Journal Requirements:

“This research was supported by the Faculty of Medicine Research Fund of Chiang Mai University (grant no. 152/2562). The funders had no role in study design, data collection and analysis, decision to publish, or preparation of the manuscript.”

“This research was supported by the Faculty of Medicine Research Fund of Chiang Mai University (grant no. 152/2562). We are thankful to our assistants for the data collection and all participants who made this research successful.”

“This research was supported by the Faculty of Medicine Research Fund of Chiang Mai University (grant no. 152/2562). The funders had no role in study design, data collection and analysis, decision to publish, or preparation of the manuscript.”

Reviewers' comments:

Reviewer's Responses to Questions

**Comments to the Author**

1. Is the manuscript technically sound, and do the data support the conclusions?

Reviewer #1: Yes

Reviewer #2: Partly

2. Has the statistical analysis been performed appropriately and rigorously? 

Reviewer #1: I Don't Know

Reviewer #2: Yes

3. Have the authors made all data underlying the findings in their manuscript fully available?

Reviewer #1: Yes

Reviewer #2: No

4. Is the manuscript presented in an intelligible fashion and written in standard English?

Reviewer #1: Yes

Reviewer #2: Yes

5. Review Comments to the Author

Reviewer #1: Dear Authors, please go through the below text for detailed concerns, each referenced with the line number in the original manuscript for your convenience.

Title: It would helpful to elaborate in your title the five precepts “of buddhism” for people not very familiar with buddhism or these precepts in the worldwide scientific community may not be clear what the manuscript is about.

Abstract: You have mentioned aim is to study how the 5 precepts affect depressive symptoms. What about other mental health symptoms? Could you mention if any studies exist looking at other mental health symptoms or diagnoses too?

These are several positives in your manuscript. You have chosen to write about and research a very interesting topic that appears to be on the cusp of neuroscience and spirituality. I appreciate that you have also explored how practicing of these precepts even in non-Buddhists may potentially be helpful, but does need to be studied separately outside the Buddhism context too, to be evaluated more universally. Since the pretext of your study is that observing the precepts may be a modulating and mediating factor in developing of depression, but it cannot be clearly shown that there is a direct causal relationship, I appreciate that you have stated this in limitations section, that the cause effect relationship cannot be confirmed as it is a cross sectional study.

48: Language could be clearer, eg. Depressive episodes often represent an interaction of the trait of neuroticism with a life stressor.

69: Abstaining from killing any living beings

102 Could you elaborate on why Exclusion criteria consisted of having psychiatric history or being treated for psychiatric disorder and having been diagnosed or being treated for substance use disorder. ?

128. Unclear as 2 Cronbach’s alpha values are mentioned in this line one was .83 and one was .90. If you can clarify which is for which.

152: Not mentioned clearly: Is SBI-FPO and SBI PP the same ?

Keywords: More specific keywords may be included, especially ones like “precepts”.

Statistics: Overall, statistical analysis appears to be done in a very rigorous manner. You have appropriately used the bootstrap method for mediation analysis, but you could also elaborate more on why this method was chosen over other analytical methods like Baron and Kenny logic or Sobel test. You have a good N of 644 which indicates good study power.

What statistical analyses have been done to rule out confounding and/or presence of third variables/confounding factors?

Cronbach’s alpha - which is a good measure for internal consistency - would be helpful to mention that it is a measure of internal consistency, whenever you mention this parameter.

Among participants most were female and lived alone - why was that the case. Any possible confounding factors related to this skewed demographic? Does this population truly represent Thai general population? Some remarks on this would help, maybe in limitations section.

Reviewer #2: Title: Adequate

Abstract: Please include from where the data was collected.

Introduction: At the end of the study, please clearly state the research gap and the need of the current study.

Material & Method: Data collection was done between the period of December 2019 and September 2020. Any reason why the data collection took 10 months?

What was the locale of the participants. Ethnicity of the participants is important for the current study because the five precepts are received and followed in different ways based on one’s cultural background.

Under exclusion criteria: substance use disorder is also one of the psychiatric disorders. So you can include it within the first exclusion criterion itself.

From page 7, citation represented by numbers are not placed as superscripts. For instance, line number 121 and 122.

Results: Please describe the sociodemographic characteristics of the participants, as well as the correlation among sociodemographic features and test scores.

Table 2: Under Marital Status, only “no partner” (unmarried?) category was given. Why other marital status categories were excluded?

Similarly, for Education and Monthly Income, not all categories are described. Please explain why.

Discussion: “Observing the Five Precepts can be trained as mindfulness mediation”. Please add citation(s) for this statement.

To justify the need and significance of the current study, please add the implications of the study at the end of the discussion.

Also, it is required to highlight the role of culture in practice of Five Precepts and how mental health professionals should be sensitive toward the same.

6. PLOS authors have the option to publish the peer review history of their article (what does this mean?). If published, this will include your full peer review and any attached files.

Reviewer #1: **Yes: **Manan J Shah

Reviewer #2: No

---

## [Author Response · Author response to Decision Letter 0]

13 Oct 2022

Department of Psychiatry, Faculty of Medicine, Chiang Mai University, Thailand

12 October 2022

Re: PONE-D-22-01483 - Moderating role of observing the five precepts on neuroticism, perceived stress, and depressive symptoms

Dear Editor,

Thank you for providing us an opportunity to revise our manuscript. We now have completed our revised manuscript based on the reviewers’ comments and suggestions. 

Please see below the point-by-point response to those comments.

“This research was supported by the Faculty of Medicine Research Fund of Chiang Mai University (grant no. 152/2562). The funders had no role in study design, data collection and analysis, decision to publish, or preparation of the manuscript.”

Please provide an amended statement that declares *all* the funding or sources of support (whether external or internal to your organization) received during this study, as detailed online in our guide for authors at http://journals.plos.org/plosone/s/submit-now. Please also include the statement “There was no additional external funding received for this study.” in your updated Funding Statement. Please include your amended Funding Statement within your cover letter. We will change the online submission form on your behalf.

Response.

Thank you very much. We have updated the cover letter and provided an amended statement that declares *all* the funding or sources of support received during this study online.

Reviewer 1

Reviewer #1: Dear Authors, please go through the below text for detailed concerns, each referenced with the line number in the original manuscript for your convenience.

Title: It would be helpful to elaborate in your title the five precepts “of buddhism” for people not very familiar with buddhism or these precepts in the worldwide scientific community may not be clear what the manuscript is about.

Response: Thank you. We agree that the title should be added “ of Buddhism” for more clarity. 

Abstract: You have mentioned aim is to study how the 5 precepts affect depressive symptoms. What about other mental health symptoms? Could you mention if any studies exist looking at other mental health symptoms or diagnoses too?

Response: We have revised this point as follows.

Evidence has shown that the Five precepts significantly affect the relationship between attachment and resilience; however, little is known whether observing the Five Precepts would help reduce depressive symptoms among those who experience risks.

These are several positives in your manuscript. You have chosen to write about and research a very interesting topic that appears to be on the cusp of neuroscience and spirituality. I appreciate that you have also explored how practicing of these precepts even in non-Buddhists may potentially be helpful, but does need to be studied separately outside the Buddhism context too, to be evaluated more universally. Since the pretext of your study is that observing the precepts may be a modulating and mediating factor in developing of depression, but it cannot be clearly shown that there is a direct causal relationship, I appreciate that you have stated this in limitations section, that the cause effect relationship cannot be confirmed as it is a cross sectional study.

Response: Thank you for this support. 

48: Language could be clearer, eg. Depressive episodes often represent an interaction of the trait of neuroticism with a life stressor.

Response. We have revised to be more clarified as follows.

A clinically significant depressive symptom is usually attributable to an interaction of the trait of neuroticism with a life stressor.

69: Abstaining from killing any living beings

Response

We revised the word ‘abstaining’ to ‘refraining’

Refraining from killing any living beings

102 Could you elaborate on why Exclusion criteria consisted of having psychiatric history or being treated for psychiatric disorder and having been diagnosed or being treated for substance use disorder?

Response.

We are afraid that having a psychiatric disorder would affect the depressive symptom and would affect how the respondent accurately responded to the questionnaire. We would like the sample to be more representative of general people who may experience some depressive symptoms but not due to the depressive disorder. 

In the case of a substance use disorder, we apologize for this mistake. We have changed “having been diagnosed or being treated for substance use disorder” to “being intoxicated”. The reason is that we cannot be confident whether they were under the influence, which would affect the reliability of the data. 

128. Unclear as 2 Cronbach’s alpha values are mentioned in this line one was .83 and one was .90. If you can clarify which is for which.

Response. We have revised it as follows.

The previous studies showed that the NI had high internal consistency (Cronbach’s alpha was .91 - .92)35,49,and NI showed good validity and reliability.48 In this study sample, the Cronbach’s alpha was .90.

152: Not mentioned clearly: Is SBI-FPO and SBI PP the same ?

Response. We apologize for this mistake, it has been changed from SBI-FPO to SBI-PP.

Keywords: More specific keywords may be included, especially ones like “precepts”.

Response: Thank you for this suggestion. We have added it accordingly.

Statistics: Overall, statistical analysis appears to be done in a very rigorous manner. You have appropriately used the bootstrap method for mediation analysis, but you could also elaborate more on why this method was chosen over other analytical methods like Baron and Kenny logic or Sobel test. You have a good N of 644 which indicates good study power.

Response. Thank you. We have added these parts shown below.

We used bootstrap instead of traditional Baron and Kenny’s mediation methods and the Sobel test because indirect effects are unlikely to be normally distributed. Bootstrap is a resampling technique with replacement, and no assumption is made about the shape of the sampling distribution of indirect effect, the results will get more credibility, and the bootstrap confidence interval tends to have higher power than the Sobel test.56

What statistical analyses have been done to rule out confounding and/or presence of third variables/confounding factors?

Response. We have added the following statements

Hierarchical regression analysis was used to identify potential confounders. The results showed that age, sex, and marital status were the significant predictors that reduced the effect size of neuroticism on depression, which was considered confounders; therefore, these three variables were controlled as covariates in the moderated mediation model.

Cronbach’s alpha - which is a good measure for internal consistency - would be helpful to mention that it is a measure of internal consistency, whenever you mention this parameter.

Response. Thank you.

We now revised it as follows.

In this study sample, the PSS-10 demonstrated a good internal consistency (Cronbach’s alpha was .78).

Among participants most were female and lived alone - why was that the case. Any possible confounding factors related to this skewed demographic? Does this population truly represent Thai general population? Some remarks on this would help, maybe in limitations section.

Response.

Yes, we agree with that. This study was limited to people who could access the online survey. The invitation was carried out using social media and flyers. It was difficult to control for equality of sex and other demographic factors. The results can only tell that females and people who live alone participated the most in the study. The disproportionate sex ratio makes it unlikely to be representative of Thai people. We have added this point to the limitation.

Reviewer #2: Title: Adequate

Response.

Thank you. However, we have added the word “of Buddhism” for clarity based on the first reviewer’s suggestion.

Abstract: Please include from where the data was collected.

Response. We have included this information.

The study employed a cross-sectional survey design, collected data from the end of 2019 to September 2022 in Thailand.

Introduction: At the end of the study, please clearly state the research gap and the need of the current study.

Response. We have added the research gap as suggested.

Little is known about the role of observance of the Five Precepts on negative mental outcomes such as perceived stress, neuroticism and depression. 

The authors therefore analyzed to see whether observance of the Five Precepts would serve as a buffer for any mental health outcome the same way as self-control does. Specifically, the authors examine the moderating effect on the relationship among neuroticism, perceived stress and depression.

Material & Method: Data collection was done between the period of December 2019 and September 2020. Any reason why the data collection took 10 months?

Response. Thank you for this careful observation.

Initially, the project provided two options for the respondent to complete the questionnaires, paper or online forms. However, after the COVID-19 pandemic began, the researchers decided to cancel the paper method to avoid human-to-human contact. As a result, the authors had to resubmit the protocol amendment to the ECs, which took time to approve before restarting data collection. 

What was the locale of the participants. Ethnicity of the participants is important for the current study because the five precepts are received and followed in different ways based on one’s cultural background.

Response. Thank you for this comment.

We have added the missing data as follows. 

All participants were Thai, and 93.3% were Buddhist.

Under exclusion criteria: substance use disorder is also one of the psychiatric disorders. So you can include it within the first exclusion criterion itself.

Response. Yes, you’re right. We apologize for this mistake. We have changed “having been diagnosed or being treated for substance use disorder” to “being intoxicated”.

From page 7, citation represented by numbers are not placed as superscripts. For instance, line number 121 and 122.

Response. Thank you for pointing these out.

We have identified lots of abnormal citations and corrected all of those.

Results: Please describe the sociodemographic characteristics of the participants, as well as the correlation among sociodemographic features and test scores.

Response. We have revised this part of the results.

Table 2: Under Marital Status, only “no partner” (unmarried?) category was given. Why other marital status categories were excluded?

Response.

The data were, in fact, not about the marital status but living status, which has two categories- alone and with partner. Therefore, we have revised these data for correctness.

Similarly, for Education and Monthly Income, not all categories are described. Please explain why.

Response. For correlational analysis, we categorized them into two groups and using point biserial correlation because it is easy for interpretation. 

Discussion: “Observing the Five Precepts can be trained as mindfulness mediation”. Please add citation(s) for this statement.

Response. We apologize for using the misleading statement.

We have revised the text as follows.

Observing the Five Precepts should be encouraged to practice as mindfulness meditation. 

To justify the need and significance of the current study, please add the implications of the study at the end of the discussion.

Also, it is required to highlight the role of culture in practice of Five Precepts and how mental health professionals should be sensitive toward the same.

Response. We have added this section as suggested. Please see below.

Implications of the study

In clinical implication, observing the Five Precepts may be promoted along with any form of mindfulness meditation or mindfulness-related therapy.[57-59] In addition to buffering adverse mental health outcomes, observing the Five Precepts has been shown to be associated with well-being.[60] Therefore, it should be promoted even among the general population and those who have yet to experience stress. Researchers should carry out the practice of the Five Precepts further in the future. For example, research concerning an association between observing the Five Precepts and other positive strengths, such as resilience, grit, perseverance, and patience, should be examined, in addition to adverse mental health outcomes. 

However, although Five Precepts can be viewed as healthy behaviours to be fostered for oneself and others, some, especially non-Buddhists, may find it uncomfortable when considering it as culture or religion related. Therefore, mental health professionals may adopt a careful approach emphasizing “behaviours” rather than religious matters, the same way mindfulness meditation is recognized. Such an approach may make it more acceptable and open to practice and further research.

Thank you again for your consideration of our revised manuscript. Hopefully, we can sufficiently address all the concerns. We are looking forward to hearing from you soon. 

Best Regards,

TW

---

## [Editor Report · Decision Letter 1]

26 Oct 2022

Moderating role of observing the five precepts of Buddhism on neuroticism, perceived stress, and depressive symptoms

PONE-D-22-01483R1

Dear Dr. Wongpakaran,

We’re pleased to inform you that your manuscript has been judged scientifically suitable for publication and will be formally accepted for publication once it meets all outstanding technical requirements.

Kind regards,

Allen Joshua George

Guest Editor

PLOS ONE

Additional Editor Comments (optional):

You have modified the manuscript incorporating all the suggestions from the reviewers. I congratulate the authors for improving the clarity and quality of the manuscript making it suitable for publication.

---

## [Editor Report · Acceptance letter]

6 Nov 2022

PONE-D-22-01483R1 

Moderating role of observing the five precepts of Buddhism on neuroticism, perceived stress, and depressive symptoms 

Dear Dr. Wongpakaran:

I'm pleased to inform you that your manuscript has been deemed suitable for publication in PLOS ONE. Congratulations! Your manuscript is now with our production department. 

Kind regards, 

on behalf of

Dr. Allen Joshua George 

Guest Editor

PLOS ONE